# An FDA-Approved Antifungal, Ketoconazole, and Its Novel Derivative Suppress tGLI1-Mediated Breast Cancer Brain Metastasis by Inhibiting the DNA-Binding Activity of Brain Metastasis-Promoting Transcription Factor tGLI1

**DOI:** 10.3390/cancers14174256

**Published:** 2022-08-31

**Authors:** Daniel Doheny, Sara Manore, Sherona R. Sirkisoon, Dongqin Zhu, Noah R. Aguayo, Alexandria Harrison, Mariana Najjar, Marlyn Anguelov, Anderson O’Brien Cox, Cristina M. Furdui, Kounosuke Watabe, Thomas Hollis, Alexandra Thomas, Roy Strowd, Hui-Wen Lo

**Affiliations:** 1Department of Cancer Biology, Wake Forest University School of Medicine, Winston-Salem, NC 27101, USA; 2Proteomics and Metabolomics Shared Resource, Comprehensive Cancer Center, Wake Forest University School of Medicine, Winston-Salem, NC 27101, USA; 3Department of Internal Medicine, Section on Molecular Medicine, Wake Forest University School of Medicine, Winston-Salem, NC 27101, USA; 4Wake Forest Baptist Comprehensive Cancer Center, Wake Forest University School of Medicine, Winston-Salem, NC 27101, USA; 5Department of Biochemistry, Wake Forest University School of Medicine, Winston-Salem, NC 27101, USA; 6Department of Hematology and Oncology, Wake Forest University School of Medicine, Winston-Salem, NC 27101, USA; 7Department of Neurology, Wake Forest University School of Medicine, Winston-Salem, NC 27101, USA

**Keywords:** tGLI1, GLI1, ketoconazole, breast cancer, cancer stem cells, brain metastasis

## Abstract

**Simple Summary:**

Breast cancer is the most commonly diagnosed cancer in American women, and metastasis accounts for the majority of breast cancer-related deaths. The common metastatic sites for breast cancer includes the bones, lungs, brain, and liver. Breast cancer brain metastasis (BCBM) patients have dismal prognoses, primarily due to the lack of understanding of the molecular mechanisms driving breast cancer cell colonization to the brain. In breast cancer, truncated glioma-associated oncogene homolog 1, tGLI1, promotes preferential metastasis to the brain through the upregulation of the cancer stem cell subpopulation and the activation of astrocytes. Whether tGLI1 is an actionable therapeutic target for any cancer type has not yet been investigated. Herein, we identified an FDA-approved antifungal, ketoconazole (KCZ), and its novel derivative, KCZ-7, to antagonize tGLI1 transcriptional activity, suppress cancer stem cells, and inhibit BCBM, rendering tGLI1, for the first time, as an actionable therapeutic target for the prevention and treatment of BCBM.

**Abstract:**

The goal of this study is to identify pharmacological inhibitors that target a recently identified novel mediator of breast cancer brain metastasis (BCBM), truncated glioma-associated oncogene homolog 1 (tGLI1). Inhibitors of tGLI1 are not yet available. To identify compounds that selectively kill tGLI1-expressing breast cancer, we screened 1527 compounds using two sets of isogenic breast cancer and brain-tropic breast cancer cell lines engineered to stably express the control, GLI1, or tGLI1 vector, and identified the FDA-approved antifungal ketoconazole (KCZ) to selectively target tGLI1-positive breast cancer cells and breast cancer stem cells, but not tGLI1-negative breast cancer and normal cells. KCZ’s effects are dependent on tGLI1. Two experimental mouse metastasis studies have demonstrated that systemic KCZ administration prevented the preferential brain metastasis of tGLI1-positive breast cancer and suppressed the progression of established tGLI1-positive BCBM without liver toxicities. We further developed six KCZ derivatives, two of which (KCZ-5 and KCZ-7) retained tGLI1-selectivity in vitro. KCZ-7 exhibited higher blood–brain barrier penetration than KCZ/KCZ-5 and more effectively reduced the BCBM frequency. In contrast, itraconazole, another FDA-approved antifungal, failed to suppress BCBM. The mechanistic studies suggest that KCZ and KCZ-7 inhibit tGLI1’s ability to bind to DNA, activate its target stemness genes *Nanog* and *OCT4,* and promote tumor proliferation and angiogenesis. Our study establishes the rationale for using KCZ and KCZ-7 for treating and preventing BCBM and identifies their mechanism of action.

## 1. Introduction

Despite improvements in early detection and targeted therapies, breast cancer remains the second leading cause of cancer-related deaths in women [1] and second most common cancer to metastasize to the brain [2]. Increasing numbers of breast cancer patients are at risk of developing breast cancer brain metastases (BCBM). Brain metastases frequently occur in metastatic breast cancer patients, with approximately 10–16% of patients developing symptomatic brain metastases and another 10% of patients noted to have asymptomatic brain involvement in post-mortem autopsies [2,3]. Breast cancer patients with brain metastases are expected to have a median survival time range of 4–6 months [4,5]. This dismal prognosis is primarily due to the inadequate understanding of the molecular pathways required for the growth of brain metastases, precluding the development of effective treatments. Patients with HER2-enriched or triple-negative breast cancers (TNBC) account for 30–50% of all invasive breast cancers and are associated with higher incidences of brain metastases compared to other breast cancer subtypes. The first-line therapy for BCBM is radiation therapy; however, recurrence is common. Most systemic therapies have limited penetrance across the blood–brain barrier (BBB) [6]. Furthermore, metastatic HER2-enriched tumors that initially respond to trastuzumab frequently acquire resistance [7], while the treatment options for metastatic TNBC remain limited, underscoring the need to establish novel actionable targets. 

Truncated glioma-associated oncogene homolog 1 (tGLI1) is an alternative splice variant of the zinc finger oncogenic transcription factor GLI1 [8]. This isoform arises from an in-frame deletion of 41 amino acids corresponding to the entirety of exon III and part of exon IV. In addition to retaining all GLI1 functional domains and acting as a terminal effector of the SHH-PTCH1-SMO signaling axis, tGLI1 has gained the ability to activate the expression of at least ten genes leading to tumor growth, angiogenesis, migration, invasion, and stemness [8,9,10,11,12,13,14,15]. In stark contrast to GLI1, tGLI1 expression is tumor-specific: tGLI1 is frequently expressed in glioblastomas, primary and metastatic breast carcinomas, and breast cancer cell lines, but is undetectable in normal mammary, brain, and other tissues [8,9,10,11,16]. Furthermore, other groups independently confirmed the roles of tGLI1 in breast cancer angiogenesis [15] and gliomas [17], and reported the role of tGLI1 in invasive hepatoma [18]. Most recently, we reported that tGLI1 is highly expressed in BCBM samples, and tGLI1-expressing breast cancer has an increased ability to undergo preferential metastasis to the brain, in part through promoting breast cancer stem cells (CSCs) and activating astrocytes in the brain microenvironment [13]. 

Given its tumor-specific expression and ability to promote BCBM, tGLI1 is an ideal therapeutic target to treat BCBM. In this study, we sought to identify tGLI1 inhibitors that could be developed further to treat tGLI1-positive BCBM. Using a dose-escalating chemical screening approach, we found that ketoconazole (KCZ), an FDA-approved imidazole antifungal, specifically killed tGLI1-expressing cells without concomitant effects on GLI1-expressing cells. KCZ was particularly efficacious against the tGLI1-positive breast CSC subpopulation, without affecting normal cells. Using experimental breast cancer metastasis mouse models, we found KCZ to effectively penetrate the BBB following systemic administration, to inhibit the progression of established tGLI1-positive BCBM, and to prevent circulating tGLI1-positive cancer cells from undergoing BCBM. Next, we modified the chemical moieties of KCZ to create six novel derivatives, and found that two (KCZ-5 and KCZ-7) retained tGLI1 selectivity in vitro, while KCZ-7 showed increased BBB penetration compared to KCZ and KCZ-5 and in vivo efficacy against tGLI1-driven BCBM. In contrast, another antifungal, and known smoothened (SMO) inhibitor [19], itraconazole (ITZ), did not inhibit BCBM in vivo. Furthermore, mechanistic studies suggest that KCZ and KCZ-7 inhibit tGLI1-positive breast cancer by reducing tGLI1 DNA binding, leading to reduced promoter occupancy and promoter transactivation and the downregulation of the tGLI1-mediated stemness genes *Nanog* and *OCT4*. Collectively, these data delineate tGLI1 as an actionable target for the treatment of breast cancer brain metastases, meriting further investigation and providing the rationale for repurposing KCZ for the prevention and treatment of BCBM. 

## 2. Materials and Methods

### 2.1. Cell Lines and Reagents

The human breast cancer cell lines MCF7, MCF10A, BT-20, and SKBR3 were purchased from ATCC (Manassas, VA, USA) and cultured as specified by ATCC. The HMLEs, a kind gift from the Weinberg laboratory, were cultured in MEBM^TM^ Mammary Epithelial Basal Medium (Lonza CC-3151) (Basel, Switzerland) supplemented with MEGM^TM^ Mammary Epithelial SingleQuots^TM^ (Lonza CC-436), 10% fetal bovine serum (FBS, Corning 35-10-CV) (Corning, NY, USA), and 1% penicillin–streptomycin solution (P/S, Corning 30-002-CI). The MDA-MB-231 and brain-metastatic MDA-MB-231BRM cell lines were from the Massagué laboratory and cultured in DMEM supplemented with 10% FBS and 1% P/S [20]. The isogenic MDA-MB-231 and MDA-MB-231BRM cell lines stably expressing the control vector, GLI1-, or tGLI1-expression vectors were established in our previous study [13]. The brain-metastatic variant of SKBR3 (SKBRM), derived from parental SKBR3 cells through three rounds of in vivo selection, was a kind gift from Drs. Fei Xing and Kounosuke Watabe [21]. The E6/E7/hTERT immortalized human astrocyte cell line, UC1, was a kind gift from Dr. Russell Pieper (University of California-San Francisco). The HBMECs were from Angio-Proteomie and were cultured in EBM™ Basal Medium (Lonza CC-3121) supplemented with EGM^TM^ Endothelial Cell Growth Medium SingleQuots^TM^ (CC-4133) with 10% FBS and 1% P/S. The HepG2 hepatocellular carcinoma cells were purchased from Sigma-Aldrich (85011430) (St. Louis, MO, USA) and cultured according to ATCC recommendations. All cell lines were authenticated using standard methods and routinely tested for mycoplasma contamination. If mycoplasma contamination was detected, the cells were treated with BM-Cyclin (Sigma-Aldrich 10-799-050-001) and tested again prior to use. Here, pCMV-Tag2b, pCMV-Tag2b-GLI1, and pCMV-Tag2b-tGLI1 plasmids were also generated in our laboratory [8]. The Nanog (HG13138-UT) and OCT4 (HG13137-UT) overexpression plasmids were purchased from Sino Biological (Beijing, China). The GLI1 and tGLI1 isogenic cell lines containing an RFP reporter were previously developed in our laboratory [8,9]. The drug libraries (L1200, L1300, L1400), KCZ (S1353), and ITZ (S2476) were purchased from Selleck Chemicals (Houston, TX, USA). The KCZ derivatives were synthesized by BioDuro (Beijing, China). See the Appendix A for details of the synthesis of KCZ derivatives and the remaining methods (Appendix A).

### 2.2. Cell-Based Chemical Screens

The isogenic MDA-MB-231 and MDA-MB-231BRM cell lines stably expressing the control vector, GLI1-, or tGLI1-expression vectors we established in our previous study [13]; MCF10A, HepG2, or immortalized human astrocytes were cultured in their respective culture media. The cells were harvested during the exponential growth phase, seeded at 2 − 4 × 10^3^ cells per well in 96-well white bottom plates (Greiner Bio-One 655083) (Kremsmünster, Austria), and incubated at 37 °C in 5% CO_2_ for 24 h. The cells were subsequently treated with the test compound or vehicle control for 48 h. The final concentration of DMSO (vehicle) was 1% for all treatments. Viability was determined using the CellTiter-Blue^®^ Cell Viability Assay (Promega G8080) (Madison, WI, USA) according to the manufacturer’s instructions.

### 2.3. Mammosphere Assay 

The adherent cells were harvested and seeded at a density of 1 − 4 × 10^3^ cells per well in 24-well ultra-low attachment plates (Corning 3473) with Dulbecco’s modified Eagle’s medium/F12 (Gibco 11320033) (Waltham, MA, USA) containing 2% B27 (Gibco 17504044), 20 ng/mL recombinant EGF (Millipore Sigma PHG0311) (Burlington, MA, USA), 4 μg/mL insulin (Millipore Sigma 12585014), and 100 ng/mL recombinant Sonic Hedgehog protein (SHH, Millipore Sigma GF174). Beginning 24 h after seeding, the mammospheres were treated with the vehicle (1% DMSO), KCZ, or KCZ derivatives. The mammospheres were cultured for 7–14 days and supplemented with 100 μL of fresh treatment prepared in mammosphere medium every 48 h. The number of spheres with a diameter of at least 100 μm was counted under 5× objective.

### 2.4. Selective Knockdown of tGLI1 Using Antisense Oligonucleotides (AS-ON) 

The control or tGLI1-specific locked nucleic acid (LNA) AS-ONs were custom designed and purchased from Qiagen (Hilden, Germany). The sequence for the negative control LNA AS-ON was /56-FAM/*A*A*C*A*C*G*T*C*T*A*T*A*C*G*C. The BLAST analysis did not show binding of the control to any gene. The sequence for the tGLI1-targeting AS-ON was /56-FAM/^+^C^+^A^+^A^+^CT*T*G*A*C*T*T*C*^+^T^+^G^+^TC. Here, phosphorothioated bases are indicated by *, whereas LNA bases are labeled by +. The knockdown of tGLI1 was conducted as described previously [11,13]. Briefly, BT-20 cells were transfected for 48 h with 100 nM control or tGLI1 AS-ON using Lipofectamine 2000 (Invitrogen 11668027) (Waltham, MA, USA). The cells were subsequently harvested for the quantitative PCR or seeded for mammosphere assays.

### 2.5. Quantitative RT-PCR

The RNeasy Mini Kit (Qiagen 74104) was used to the isolate total RNA and the cDNA was produced from 1 μg of total RNA using the Superscript III First-Strand cDNA synthesis system (Invitrogen 18080044). The quantitative PCR was carried out as previously described [11,12], using the primers described in Appendix A.

### 2.6. Animal Studies

Female nude mice of 6–7 weeks of age (Charles River, Wilmington, MA, USA) were housed in a pathogen-free facility at the Animal Research Program at Wake Forest School of Medicine (WFSM) under a 12/12 h light/dark cycle and fed irradiated rodent chow ad libitum. The animal handling procedures were approved by the WFSM Institutional Animal Care and Use Committee (IACUC). In the tumor prevention model, the mice received a single 100 μL intraperitoneal treatment of either the vehicle or 50 mg/kg KCZ dissolved in 100% polyethylene glycol 300 (PEG-300, Sigma 202371) 24 h prior to the intracardiac inoculation with 2 × 10^5^ exponentially growing SKBRM-tGLI1 cells in 100 μL ice-cold PBS. Successful inoculations were confirmed via the visualization of brain bioluminescent signals within 60 min following inoculation; otherwise, the mice were immediately sacrificed. The tumor progression was monitored with biweekly bioluminescent imaging (BLI), in which xenograft-bearing mice were intraperitoneally injected with 100 mg/kg d-luciferin (Perkin Elmer 122799) (Waltham, MA, USA) and imaged using the IVIS Lumina LT Series III imager (Perkin Elmer). The treatments were administered three times per week until study termination. The tumor burden was analyzed by quantifying the BLI signal in each region-of-interest measured in total flux (p/s) with the Living Image software version 4.7.2 (Perkin Elmer). For the tumor treatment model, the mice were intracardially inoculated with 2 × 10^5^ exponentially growing SKBRM-GLI1 or SKBRM-tGLI1 cells. The successfully inoculated mice were randomized into vehicle or drug treatment groups (50 mg/kg KCZ-5, KCZ-7, or KCZ dissolved in PEG-300; 50 mg/kg ITZ dissolved in 10% *N*,*N*-Dimethylacetamide (DMAc, Sigma ARK2190) and 90% PEG-300), with treatment beginning 13 days after inoculation. The tumor growth was monitored as described for the tumor prevention model.

### 2.7. Production of tGLI1 Recombinant Protein

The full-length tGLI1 coding sequence was cloned into a modified pET28 expression vector (pLM303-tGLI1) containing an N-terminal maltose-binding protein (MBP) tag and an intervening rhinovirus 3C protease cleavage site. The expression construct was transformed into BL21(DE3)-competent *E. coli* (Sigma CMC0014) and grown in Luria–Bertani (LB) medium at 37 °C with shaking to an OD_600_ = 0.6. The cultures were induced with 0.3 mM isopropyl-β-D-thiogalactoside (IPTG) and allowed to express protein at 16 °C overnight. The harvested bacteria were resuspended in bacterial resuspension buffer (50 mM Tris pH 7.5, 300 mM NaCl, 1 mM MgCl_2_, 0.1 mM EDTA, 10% glycerol) and cOmplete protease inhibitor cocktail (Roche, Penzberg, Germany), then lysed using an Avestin Emulsiflex-C5 cell homogenizer. The cell debris was cleared via centrifugation at 10,000 rpm at 4 °C for 30 min, and the cleared lysate was passed over amylose high-flow resin (New England Biolabs, Ipswich, MA, USA) and washed with at least 3 column volumes of amylose column buffer (ACB) (50 mM Tris pH 7.5, 250 mM NaCl, 2 mM MgCl_2_, 0.1 mM EDTA, 2 mM DTT, 10% glycerol). The bound MBP-tGLI1 protein was eluted with ACB plus 20 mM maltose. The desired fractions were pooled and dialyzed overnight against the heparin column buffer (HCB) (50 mM Tris pH 7.5, 100 mM NaCl, 2 mM MgCl_2_, 0.1 mM EDTA, 2 mM DTT, 5% glycerol) containing HRV 3C PreScission Protease (GE Biosciences, Chicago, IL, USA) to cleave the MBP tag. The tGLI1 recombinant protein was separated from the cleaved MBP using a Heparin HiTrap (GE Healthcare Life Sciences, Chicago IL, USA) and eluted using a linear gradient of HCB plus 2 M NaCl. The fractions containing tGLI1 were pooled and spin-concentrated using a Vivaspin 20 instrument (Vivaproducts VS2002) (Littleton, MA, USA), then the aliquots were frozen on dry ice and stored at −80 °C until use.

### 2.8. Electrophoretic Mobility Shift Assay

Approximately 600 ng of recombinant STAT3 (Creative Biomart, STAT3-29823TH) (Shirley, NY, USA), GLI1 (Creative Biomart, GLI1-312H), or N-tGLI1 protein was mixed with 5X binding buffer (50 mM Tris pH 7.5, 50 mM NaCl, 200 mM KCl, 5 mM MgCl_2_, 10 mM EDTA, 5 mM DTT, 250 μg/mL BSA, 25% glycerol), 50 ng/μL poly dI·dC (Sigma P4929), and 5 pmol 6FAM-labeled dsDNA oligo (Integrated DNA Technologies, Coralville, IA, USA) in a total reaction volume of 20 μL. The oligos were ordered as the dsDNA from IDT with the sequences /56-FAM/CGAAGAGACCACCCAGGTAGCT and /56-FAM/AGCTACCTGGGTGGTCTCTTCG; the GLI1 consensus binding sequence is underlined. The binding reactions were incubated on ice for 20 min before electrophoresis on 6% (19:1) acrylamide–bisacrylamide TBE gels using a 0.5× TBE + 2.5% glycerol running buffer at 80 V. For the drug disruption studies, the binding buffer, protein, and treatment solutions were combined and incubated for 30 min on ice or at room temperature for GLI1 or N-tGLI1, respectively. The final treatment concentrations were 1% DMSO, 100 μM KCZ, or 100 μM KCZ-7. After the addition of the dsDNA oligo and poly dI·dC, the reactions were incubated for an additional 30 min before electrophoresis. The gels were imaged using the fluorescein module on a ChemiDoc MP system (BioRad, Hercules, CA, USA).

### 2.9. Chromatin Immunoprecipitation

The SKBRM cells in the exponential growth phase were transfected with GLI1 or tGLI1 expression plasmid and a GLI1 binding site-driven luciferase construct 8 × 3′GLI1 generously provided by Dr. Hiroshi Sasaki (Osaka University) [22]. After 24 h, the cells were treated with 1% DMSO, KCZ, or KCZ-7 for 20 h. The cells were stimulated with 100 ng/mL SHH for 4 h prior to crosslinking with 1% formaldehyde. The excess formaldehyde was quenched with 0.125 M glycine and the ChIP assay was carried out as described previously using the ChIP assay kit from EMD Millipore (Cat No. 17-371) [11]. The GLI1 and tGLI1 cell lysates were immunoprecipitated using a GLI1 antibody (Cell Signaling Technology/CST 2643) (Danvers, MA, USA) that recognizes both GLI1 and tGLI1 and has been successfully used for ChIP [23]. The normal mouse IgG served as the negative immunoprecipitation control and the input chromatin was used as the loading control for the quantitative RT-PCR. The primers used for the detection of the GLI1-binding site were 5′-GAGTCAGTGAGCGAGGAAG-3′ and 5′-GCCGGGCCTTTCTTTATGT-3′. 

### 2.10. Western Blotting

The immunoblotting was performed as previously described [11,13,24]. The antibodies included GLI1 (CST; 2643, 1:1000), a custom-made tGLI1-specific antibody (Yenzyme, 1:1000), OCT4 (CST 2750 and CST 4286, 1:1000), Nanog (CST 4903, 1:1000), androgen receptor (CST 5153, 1:1000), α-Tubulin (Sigma T6074, 1:5000), β-actin (CST 3700, 1:5000), and Vinculin (CST 13901, 1:5000). The densitometry was performed using ImageJ v1.53c (NIH, Bethesda, MD, USA).

### 2.11. Promoter Reporter Assay

The SKBR3 cells in the exponential growth phase were transfected with Vector, GLI1, or tGLI1 expression plasmid; 8 × 3′GLI1-luciferase reporter; and Renilla luciferase (pRL-TK) to control for the transfection efficiency for firefly luciferase using XtremeGene HP (Roche). After 24 h, the cells were treated with the vehicle (1% DMSO) or increasing doses of KCZ or KCZ-7 for 20 h. The cells were stimulated with 100 ng/mL SHH for 4 h before the cell lysates were harvested and the luciferase activity was measured using a Firefly and Renilla luciferase kit (Biotium 30081) (Fremont, CA, USA) on a SpectraMax iD3 plate reader (Molecular Devices, San Jose, CA, USA). The total transfection and drug treatment times were 48 and 24 h, respectively. The relative promoter activity was determined by normalizing the luciferase activity to the Renilla control.

### 2.12. Statistical Analysis

The data were analyzed and graphed using Prism 9.1 (GraphPad, San Diego, CA, USA). The descriptive statistics are presented as means ± SEM. The repeated-measures ANOVAs with post hoc Bonferroni or Dunnett’s multiple comparison test were performed using Prism 9.1 and used to analyze differential drug effects in the chemical screen, colony formation, mammosphere formation, and gene expression assays. The Student’s *t*-tests, one-way ANOVAs with Dunnett’s or Tukey’s multiple comparison tests, and nonlinear regression analyses were also performed using Prism.

## 3. Results

### 3.1. KCZ Selectively Inhibits Breast Cancer Cells Expressing tGLI1 and Displays Increased Potency against the CSC Population

To identify potential tGLI1 inhibitors, we conducted a cell-based chemical screen including 1527 compounds (1504 different compounds after excluding overlapping compounds among different libraries) from three commercial libraries (Selleck Chemicals) using isogenic TNBC MDA-MB-231 and MDA-MB-231BRM lines engineered to express an empty vector, GLI1-, or tGLI1 expression vector (Figure 1a). In the initial screen, the viability was assessed using the CellTiter-Blue^®^ Viability Assay and tGLI1-selective compounds were defined as having reduced viability by 20% at 5 μM in the tGLI1-expressing cells compared to the other two lines. The results (Appendix A) showed 10 compounds with tGLI1 selectivity for MDA-MB-231 and 10 compounds with tGLI1 selectivity for MDA-MB-231BRM. KCZ was the only tGLI1-selective compound for both lines. Consequently, KCZ was further tested in a dose-escalating screen that included an expanded dose curve spanning 1 nM to 10 μM to reduce the rate of false positives (Type I error). Using this approach, we found that KCZ selectively inhibited tGLI1-expressing, but not the control vector or GLI1-expressing, MDA-MB-231 (EC_50_ = 354.3 pM) and MDA-MB-231BRM (EC_50_ = 377.9 pM) cell lines (Figure 1b). 

Given tGLI1’s role in promoting breast CSCs [13]—a small population of cells thought to be responsible for cancer progression, metastasis, and recurrence—we wanted to determine whether KCZ could target this subpopulation. Since MDA-MB-231 and MDA-MB-231BRM cells are unable to reliably form mammospheres, which are commonly used to model the breast CSC population [25], a colony formation assay was performed instead. Briefly, the cells were seeded at a low density and treated with the vehicle or KCZ for 10 days, after which the colonies were counted following Crystal Violet staining. The treatment with KCZ significantly reduced the colony formation of tGLI1-expressing MDA-MB-231 (EC_50_ = 118.7 pM) and MDA-MB-231BRM (EC_50_ = 417.4 pM) relative to both the vector and GLI1-expressing cells, suggesting that KCZ targets the CSC subpopulation (Figure 1c). In contrast, the colony formation of the vector and GLI1-expressing cell lines was unaffected by the KCZ treatment (Figure 1c). Given the pronounced effect of KCZ on the tGLI1-positive brain metastatic breast CSC subpopulation, we then assessed the effect of KCZ on normal cells residing in brain and breast microenvironments. Interestingly, KCZ did not significantly affect the growth of any of the four cell lines tested, which is attributed to the lack of endogenous tGLI1 protein expression (Figure 1d). KCZ only induced significant toxicity at the 10 μM dose in immortalized human astrocytes (Figure 1d, top left), while a normal human brain endothelial cell line (HBMEC; Figure 1d, bottom left) and two immortalized human mammary epithelial cell lines (HMLE and MCF10A: Figure 1d, top right; Figure 1d, bottom right, respectively) were unaffected by KCZ. These findings suggest that KCZ selectively inhibits tGLI1-expressing breast cancer cells and does not significantly impact GLI1-expressing breast cancer cells or normal cells found in the brain and breast microenvironments.

### 3.2. tGLI1 Expression Is Required for the KCZ-Induced Suppression of Breast CSCs

To further investigate the ability of KCZ to inhibit the breast CSC subpopulation, we performed mammosphere formation assays. In agreement with the colony formation assay (Figure 1c), KCZ significantly reduced the mammosphere formation of HER2-enriched tGLI1-expressing brain metastatic SKBR3 (SKBRM) cells (EC_50_ = 15.06 pM) without concomitant effects on the vector control (EC_50_ = 461.1 nM) or GLI1-expressing (EC_50_ = 573.2 nM) mammosphere formation (Figure 2a). We have previously shown that tGLI1 expression is induced under mammosphere-forming conditions compared to the standard monolayer culture [13]. Given this, we wanted to compare the ability of KCZ to inhibit the CSC subpopulation compared to the total cell population. BT-20 and MCF7 cells were concurrently grown as either monolayer (ML) or mammospheres (MS) and treated with KCZ. KCZ significantly suppressed mammosphere formation, but not monolayer viability, in both the BT-20 (EC_50_ = 68.17 pM) and MCF7 (EC_50_ = 122.6 pM) cells, suggesting that KCZ effectively targets the breast CSC subpopulation expressing endogenous levels of tGLI1 (Figure 2b). The qPCR analysis confirmed that tGLI1 expression was significantly enriched in mammospheres compared to the monolayer in both BT-20 and MCF7 cell lines (Figure 2c). Next, we investigated whether tGLI1 knockdown would abolish the KCZ-sensitivity of CSCs. Briefly, BT-20 cells were transfected with either a non-targeting locked nucleic acid (LNA) anti-sense oligonucleotide (AS-ON) or a tGLI1 targeting AS-ON before seeding in mammosphere-forming conditions. Phosphorothioated and LNA bases of the oligonucleotide precluded the nuclease-mediated degradation. The transfection with the tGLI1 AS-ON specifically reduced tGLI1, but not GLI1, mRNA expression and reduced tGLI1 protein expression, confirming the specificity (Figure 2d,e). While transfection with the non-targeting AS-ON did not significantly affect the BT-20 mammosphere formation, the tGLI1 knockdown significantly reduced the average number of mammospheres (Figure 2f), confirming the previous results that tGLI1 promotes mammosphere formation [13]. Importantly, the tGLI1 knockdown abolished KCZ’s effects on mammosphere formation, demonstrating the requirement of tGLI1 in the KCZ-mediated suppression of CSCs (Figure 2f). Collectively, these results demonstrate that tGLI1 is required for the ability of KCZ to inhibit the breast CSC subpopulation.

### 3.3. KCZ Reduces the Ability of tGLI1-Positive Circulating Breast Cancer Cells to Undergo Colonization and Form Brain Metastases In Vivo

Given the previous evidence implicating tGLI1 in breast cancer metastasis brain-tropism [13] and the heightened KCZ sensitivity of tGLI1-positive breast CSCs in vitro (Figure 2 and Figure 3), we next investigated whether KCZ could inhibit the ability of tGLI1-positive circulating breast cancer cells to undergo BCBM in vivo by treating mice with the vehicle or KCZ (50 mg/kg, ip) 24 h prior to the intracardiac implantation of SKBRM-tGLI1 cells, then continuing the KCZ treatment (50 mg/kg, ip, 3 times/week) through the end of the study (Figure 3a). The KCZ treatment significantly impaired the development of brain, but not bone, metastases compared to the control (Figure 3b,d). Additionally, the lung metastasis size also significantly decreased with the KCZ treatment (Appendix A). Ex vivo imaging of the resected samples showed that KCZ reduced the frequency of brain and bone metastases by 45% and 11%, respectively (Figure 3c). Furthermore, the KCZ treatment was well tolerated by the mice as the body weight showed no significant differences compared to the vehicle throughout the study (Figure 3e). Together, the results in Figure 3 show that KCZ reduces the ability of circulating tGLI1-positive breast cancer cells to undergo colonization and form brain metastases in vivo.

### 3.4. KCZ Selectively Inhibits the Progression of Established tGLI1-Positive Breast Cancer Brain Metastases In Vivo

Despite observing a significant reduction in brain metastasis size with KCZ treatment in the previous BCBM prevention model (Figure 3), we could not conclude that KCZ penetrates the BBB to inhibit established breast cancer brain metastases. To address this, we conducted a second in vivo study in which the KCZ treatment (50 mg/kg, ip) began upon the detection of brain metastases 13 days following intracardiac inoculation with either GLI1-expressing or tGLI1-expressing SKBRM cell lines (Figure 4a). The ex vivo bioluminescent analysis revealed a significant reduction in tGLI1, not GLI1, brain metastasis size with KCZ treatment (Figure 4b). Furthermore, the untreated tGLI1 brain metastases were significantly larger than untreated GLI1 metastases (Figure 4b), confirming the previous results demonstrating tGLI1’s role in promoting the aggressiveness of breast cancer brain metastases [13]. The lung metastasis size was not significantly affected by the KCZ treatment in either group (Appendix A). The KCZ treatment reduced the frequency of detected SKBRM-tGLI1 brain metastases by 38% but did not affect the SKBRM-GLI1 group (Figure 4c). Neither the bone nor lung metastasis frequency was reduced with the KCZ treatment (Appendix A). Concordant with the BCBM prevention study (Figure 3), the bone metastasis size was not significantly reduced by the KCZ treatment (Figure 4d). The KCZ treatment was also well tolerated in this study (Figure 4e). There was no significant increase in serum alanine transaminase (ALT) activity, a marker of acute liver damage, with the KCZ treatment in mice with either GLI1- or tGLI1-expressing metastases (Figure 4f). Of note, the measured ALT activity was well below the published threshold values (109 ± 18 U/L) using thioacetamide, which is commonly used to induce acute liver damage [26]. Taken together, these data demonstrate KCZ’s selectivity and efficacy against the progression of established tGLI1-positive brain metastases in vivo.

### 3.5. Novel KCZ Derivative KCZ-7 Retains tGLI1 Selectivity and In Vivo Efficacy While Readily Permeating the BBB

Given the experimental evidence demonstrating the ability of KCZ to selectively inhibit the progression of tGLI1-positive BCBM in vivo and the potential hepatotoxicity associated with KCZ use in humans [27], we sought to determine if alterations of the chemical moieties of KCZ could increase the BBB penetration and reduce the liver damage while retaining the tGLI1 selectivity. To this end, we designed six novel KCZ derivatives by performing single moiety substitutions (Appendix A). Viability assays using HepG2 hepatocellular carcinoma cells to model the liver metabolism, human mammary epithelial cells, and human astrocytes were performed to evaluate the off-target toxicity. KCZ-4, KCZ-6, and KCZ-10 elicited low-dose toxicity in HepG2 cells, while no compound induced significant toxicity in either human mammary epithelial cells or human astrocytes (Appendix A–d). The tGLI1 selectivity of each derivative was assessed using the SKBRM mammosphere and cell viability assay. Of the six novel derivatives, KCZ-5 and KCZ-7 demonstrated tGLI1 selectivity and potency rates similar to KCZ in the SKBRM mammosphere assay (Figure 5a, Appendix A). We then compared the bioavailability and BBB penetrance of these derivatives to the parent compound in tumor-naïve mice. The mass spectrometry analysis showed that the concentration of KCZ-7 was significantly higher than those of KCZ and KCZ-5 in the brain tissue of mice treated with a single intraperitoneal 50 mg/kg dose, while its serum concentration was not significantly different from that of KCZ (Figure 5b). Given the higher BBB penetrance of KCZ-7, we next tested this compound in the BCBM tumor treatment mouse model (Figure 5c). The treatment with KCZ-7 significantly reduced the size (Figure 5d) and detection frequency (Figure 5e) of established SKBRM-tGLI1 brain metastases compared to the control treatment, while the bone metastases (Figure 5f) and lung metastases (Appendix A) were unaffected. The size of the KCZ-7-treated brain metastases trended toward being significantly smaller compared to those treated with KCZ; however, this comparison did not reach significance (*p* = 0.079) (Figure 5d). Both agents were well tolerated over the course of the study (Figure 5g) and the serum ALT levels were not significantly increased with treatment with either KCZ or KCZ-7 (Figure 5h). To confirm the effects of KCZ and KCZ-7 on BCBM in vivo, we performed an immunohistochemistry (IHC) analysis of resected brain samples. The SKBRM-tGLI1 BCBM treated with KCZ or KCZ-7 presented with significantly reduced tGLI1 protein expression, proliferative index (Ki-67), VEGF-A protein expression, and microvessel density (mCD31) levels (Figure 5i–m). Interestingly, the FDA-approved antifungal itraconazole (ITZ), a broad-spectrum triazole developed by Jansen Pharmaceutica that gradually replaced KCZ in the 1990s, did not significantly reduce the size of tGLI1-positive brain, bone, or lung (Appendix A–d) metastases, despite being a known SMO inhibitor [19]. ITZ also did not reduce the frequency of detected brain metastases at the study endpoint (Appendix A). These treatments were also well tolerated, as indicated by the lack of significant differences in body weight or serum ALT activity (Appendix A). These data demonstrate that KCZ-7 retains its tGLI1 selectivity and in vivo efficacy and displays enhanced BBB permeability. 

### 3.6. KCZ and the Novel Derivative KCZ-7 Inhibit tGLI1’s DNA-Binding Activity, Leading to Reduced Expression of tGLI1-Targeted Stemness Genes Nanog and OCT4

Next, we aimed to uncover the mechanism by which KCZ and KCZ-7 inhibit tGLI1-positive breast cancer. First, we asked if KCZ or KCZ-7 inhibited tGLI1-positive breast cancer through androgen receptor antagonism, since KCZ is known to inhibit the androgen receptor and has shown clinical efficacy as a second-line therapy for castration-resistant prostate cancer [28]. The Western blots illustrated that the isogenic SKBRM cell lines used in the in vivo and in vivo models did not express the androgen receptor to an appreciable extent, precluding KCZ and KCZ-7 from inhibiting tGLI1-positive breast cancer through androgen receptor antagonism (Appendix A). Second, we asked whether KCZ reduced tGLI1 protein expression. Treatment with 1 μM KCZ or KCZ-7 for 24 h did not reduce tGLI1 protein expression in isogenic SKBRM (Figure 6a) or MDA-MB-231 (Appendix A) cell lines. Next, we hypothesized that these compounds disrupt tGLI1’s transcriptional activity to exhibit efficacy against tGLI1-positive breast cancer cells. We began by expressing and purifying recombinant tGLI1 protein because it is not commercially available. Despite only being 40 kDa, the recombinant protein was detected by our custom-made tGLI1 antibody that recognizes the N-terminal junction region specific to tGLI1 (Figure 6b, left; Appendix A) [12]. The sequencing by mass spectrometry confirmed that this N-terminal tGLI1 (N-tGLI1) protein contains all 5 zinc finger domains found in full-length GLI1 and tGLI1 (Appendix A) [8]. The electrophoretic mobility shit assays (EMSA) confirmed the ability of the N-tGLI1 protein to bind the consensus GLI1/tGLI1-binding sequence (Figure 6b, right). The recombinant GLI1 (commercially available) bound to the probe, as expected, whereas the recombinant STAT3 (negative control) did not bind to the probe. Importantly, both KCZ and KCZ-7 selectively reduced the binding of N-tGLI1, but not GLI1, to the consensus GLI1/tGLI1-binding sequence (Figure 6c), suggesting that KCZ and KCZ-7 reduce the tGLI1 activity by reducing the tGLI1 DNA binding. To complement this finding, we performed chromatin immunoprecipitation (ChIP) using SKBRM cells and found that the KCZ and KCZ-7 treatments reduced the binding of tGLI1, but not GLI1, to the GLI1 promoter (Figure 6d). Finally, we investigated whether KCZ or KCZ-7 could reduce the tGLI1-mediated gene transcription, since tGLI1 transactivates gene promoters containing the GLI1 consensus binding site to upregulate at least ten target genes not regulated by GLI1 [8,9,10,11,13]. The luciferase assays demonstrated that the tGLI1-mediated transactivation of the GLI1-binding site was more sensitive to the KCZ and KCZ-7 treatments compared to GLI1-mediated promoter transactivation (Figure 6e,f). The gene expression analysis revealed that the tGLI1-mediated stemness genes *Nanog* and *OCT4* were significantly downregulated in tGLI1-expressing SKBRM (Figure 6g,h; Appendix A) and MDA-MB-231 (Appendix A) cells following treatment with KCZ or KCZ-7. The Western blot analysis confirmed the downregulation of Nanog and OCT4 protein expression following treatment with KCZ or KCZ-7 (Figure 6i, Appendix A). Moreover, the overexpression of either Nanog or OCT4 rescued the SKBRM-tGLI1 mammospheres from the KCZ and KCZ-7 treatments (Figure 6j,k; Appendix A), indicating that the downregulation of these stemness genes is critical for KCZ and KCZ-7’s effects on tGLI1-positive breast CSCs and brain metastases. Together, the results in Figure 6 show that KCZ and KCZ-7 inhibit tGLI1’s DNA-binding activity, leading to the reduced expression of the tGLI1-targeted stemness genes *Nanog* and *OCT4*, leading to reduced mammosphere formation.

## 4. Discussion

We previously published the discovery of the novel gain-of-function GLI1 transcription factor, tGLI1, and its role in breast cancer progression and metastasis [8,9,10,11,12,13,24]. Importantly, we also reported that tGLI1 promotes breast CSCs and breast cancer metastasis to the brain [13]. Given tGLI1’s tumor-specific expression and potent metastasis-promoting effects, tGLI1 is an ideal therapeutic target, and we endeavored to identify a tGLI1 inhibitor to treat tGLI1-positive breast cancer. We made the following significant novel observations in this study: (a) the systematic dose escalation screening revealed the KCZ, an FDA-approved imidazole antifungal, selectively inhibits tGLI1-expressing, but not GLI1-expressing, breast cancer cells with increased efficacy against the breast CSC subpopulation in vitro; (b) tGLI1 expression is required for KCZ-mediated breast CSC inhibition; (c) the systemic KCZ treatment significantly inhibited the tGLI1-positive BCBM engraftment and progression; (d) the novel KCZ derivative KCZ-7 retains tGLI1 selectivity in vitro while demonstrating increased BBB penetrance; (e) KCZ and KCZ-7 inhibit tGLI1-positive BCBM, in part, through reducing tGLI1 DNA binding, leading to the suppression of tGLI1-mediated promoter transactivation and the downregulation of the tGLI1-mediated stemness genes *Nanog* and *OCT4*; (f) in contrast, itraconazole, another FDA-approved antifungal, failed to suppress BCBM. By reporting these findings, our study validates tGLI1 as a novel actionable target for the treatment of breast cancer brain metastases. 

Drug repurposing, in which previously developed drugs are applied to new indications, helps bypass the traditional drug development pipeline for areas of unmet clinical need [29,30]. The anticancer potential of azole antifungal drugs has been considerably investigated since KCZ was repurposed as a treatment option for hormone-dependent prostate cancer due to its anti-steroidogenesis activity [28]. However, in clinical breast cancer trials, KCZ is typically used to prevent the clearance of the primary therapeutic through CYP3A4 modulation [31]. Furthermore, the BBB penetrance of KCZ in the context of brain metastases remains largely unknown. To this end, we initiated a window-of-opportunity study in patients with BCBM and recurrent gliomas to determine if KCZ crosses the BBB and alters tGLI1 signaling in humans (NCT03796273). 

Other researchers have reported the anticancer activity of several azole antifungal drugs, notably clotrimazole [32], KCZ [33,34], and ITZ [19,35], in several cancer cell lines. However, with the exception of KCZ, no other azole antifungal was active in our tGLI1 selectivity cell-based chemical screen. Interestingly, ITZ inhibits the Hedgehog (Hh) signaling pathway [19], of which tGLI1 is a terminal effector, by antagonizing SMO activity. The lack of efficacy of ITZ in our in vivo studies (Appendix A) suggests that tGLI1 can be activated by non-canonical pathways or through crosstalk with other oncogenic pathways including TGF-β, epidermal growth factor receptor (EGFR), Notch, and Wnt/β-catenin [36,37]. In patients with TNBC, the co-activation of the Hh and Wnt signaling pathways is associated with an inferior prognosis and greater risk of tumor recurrence [38]. The transcriptome analysis of WNT3a-responsive TNBC cell lines revealed multiple Wnt target genes that are involved in Hh pathway signaling [39]. Additional research by Maeda et al. suggested that Wnt signaling via the downstream activation of β-catenin may increase the transcriptional activity of GLI1 [40]. GSK3β, a negative regulator of the Wnt pathway, is able to directly bind and phosphorylate SUFU, leading to the release of GLI1, suggesting that GSK3β may act as a positive regulator of the Hh pathway [41]. The ability of the Wnt pathway to promote GLI1, and potentially tGLI1, activation in combination with the limited efficacy of SMO inhibitors in breast cancer clinical trials provides the rationale for directly targeting tGLI1 as a potential therapeutic modality. 

Our results from two mouse models of breast cancer metastasis indicate that KCZ and the novel derivative KCZ-7 retain tGLI1 selectivity and antitumor efficacy in vivo. Despite the apparent increase in BBB permeability of KCZ-7 compared to KCZ (Figure 5b), there was not a significant increase in efficacy in vivo (Figure 5d). The addition of a toluene moiety to the terminal acetyl group of KCZ to create KCZ-7 decreased the compound polarity and likely increased the passive diffusion across the BBB [6,42]. However, this minor substitution was not sufficient to increase the efficacy against tGLI1-positive BCBM. The drug concentration at the target site is not the sole factor that influences the therapeutic efficacy in the central nervous system (CNS) [43]. Rather, the thermodynamics and kinetics of drug target binding, e.g., the rates of drug target complex formation (*k*_on_), breakdown (*k*_off_), and drug target residence times (1/*k*_off_), likely play dominant roles in determining target engagement in this protected environment, given that the drug exposure in the CNS is lower than in the systemic circulation. Therefore, the toluene moiety was likely insufficient to significantly improve the drug target binding and further reduce the tGLI1 activity. 

The crystal structures of the full-length GLI1 and tGLI1 proteins have not been solved yet. Even though the recombinant N-tGLI1 protein used in these studies contained the DNA binding domain (Figure 6b), the BL21(DE3) system was only capable of producing the first ~40 kDa of the tGLI1 protein (Appendix A). The full-length GLI1 structure has not been reported due to its large size, poor solubility, and high level of disorder, which likely underlies the difficulty we experienced in attempting to purify full-length tGLI1. Therefore, it is an important future task to synthesize the full-length tGLI1 protein. It is an equally important task to resolve the structures of both proteins to facilitate further development efforts. Solving the tGLI1 protein structure and the identification of the core chemical structures required to inhibit tGLI1 activity will be essential to enhancing the inhibitor selectivity, efficacy, blood–brain barrier penetrance, and bioavailability. Furthermore, the creation and characterization of KCZ-7 provides evidence that further investigation of the structure–activity relationship between azoles and tGLI1 antagonism is merited. 

In this study, we demonstrated that KCZ and KCZ-7 inhibit tGLI1-positive breast cancer through the antagonism of tGLI1 transcriptional activity. A previous Kaplan–Meier analysis of Gene Expression Omnibus (GEO) datasets indicated that breast cancer patients with high *OCT4* expression, but not *Nanog* expression, had shortened brain-metastasis-free survival times, pointing to the clinical utility of reducing *OCT4* expression through tGLI1 inhibition to treat patients with BCBM [13]. Bae et al. recently demonstrated that KCZ inhibits the proliferation and motility of MCF7 and MDA-MB-231 breast cancer cells via the induction of G_1_-phase arrest, and also observed reduced invasiveness through the inhibition of matrix metalloproteinase 9 (MMP9) in MDA-MB-231 but not MCF7 cells [34]. Interestingly, these results were observed irrespective of the tGLI1 status. It stands to reason that given the more aggressive phenotype, MDA-MB-231 cells exhibit a higher basal expression level of tGLI1 relative to MCF7 cells, which may, in part, explain the lack of MMP9 inhibition in MCF7 cells treated with KCZ. We previously reported that tGLI1 modulates the invasion of glioblastomas and breast cancer through the upregulation of heparanase [12] and MMP9 [9] expression, respectively. In light of the findings published by Bae et al., it will be important to investigate the inhibition of breast cancer invasion and cell cycle arrest by KCZ and KCZ-7 in the context of the tGLI1 status to fully characterize the mechanism of these compounds. 

## 5. Conclusions

In summary, our findings demonstrate that tGLI1 is an actionable target for the prevention and treatment of breast cancer brain metastases. These results, in concert with previous studies, strengthen the rationale to further investigate: (a) KCZ and other azole compounds as candidate cancer therapeutics; (b) the non-canonical pathways leading to GLI1 and tGLI1 activation; (c) the structure–activity relationship between azole compounds and tGLI1 antagonism; and (d) additional mechanisms by which KCZ and related compounds inhibit tGLI1-positive breast cancer.

## Figures and Tables

**Figure 1 cancers-14-04256-f001:**
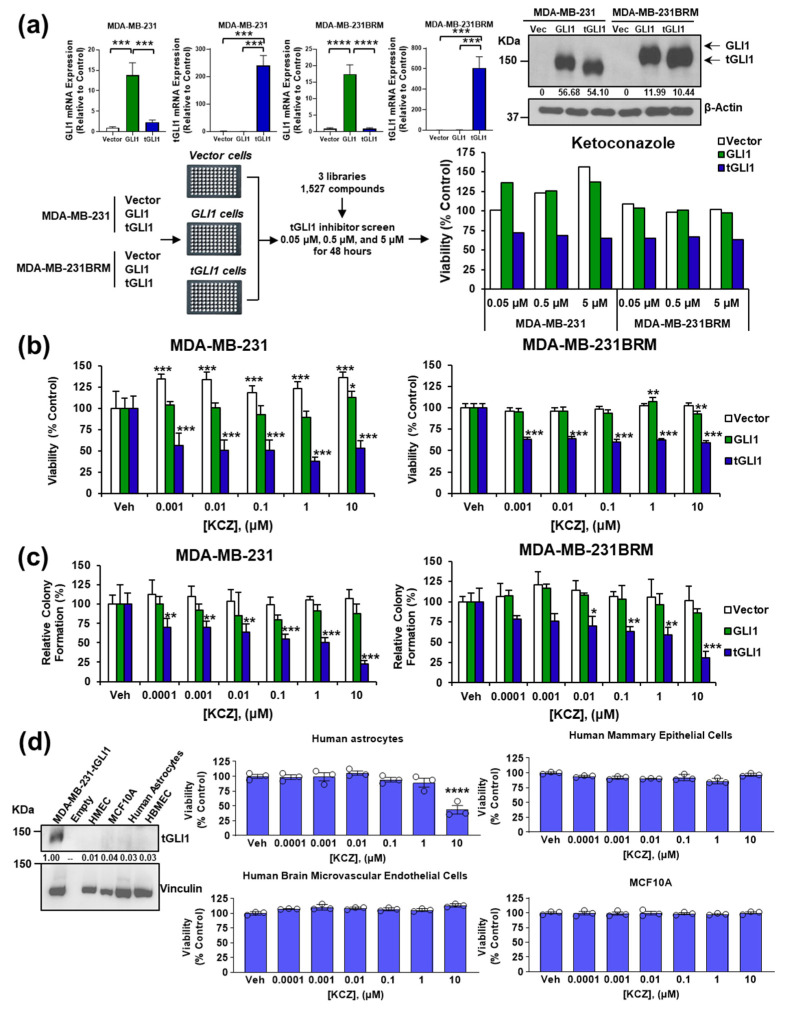
KCZ selectively inhibits breast cancer cells expressing tGLI1 with increased potency against the CSC population. (**a**) Confirmation of MDA-MB-231 and MDA-MB-231BRM cell lines stably expressing control vector, GLI1-, or tGLI1-expression vectors using RT-qPCR and a Western blot analysis. Schematic summarizing the first phase of the tGLI1 inhibitor screen that flagged KCZ as a potential tGLI1 inhibitor. MDA-MB-231 and MDA-MB-231BRM cell lines stably expressing the control vector, GLI1-, or tGLI1 expression vectors were seeded in 96-well plates and treated with the test compound. (**b**) KCZ’s effects were confirmed by a dose-escalating screen to confirm tGLI1 selectivity. Asterisks denote results of intra cell line post hoc Dunnett’s multiple comparison test against vehicle treatment. (**c**) Colony formation assay in which MDA-MB-231 and MDA-MB-231BRM cells stably expressing either an empty vector, GLI1, or tGLI1 were seeded at a low density (250 cells/well) and treated for 10 days. Asterisks denote results of post hoc Dunnett’s multiple comparison test against vehicle treatment. (**d**) The tGLI1 protein expression in normal brain and breast microenvironmental cells using a Western blot analysis. The positive control is represented by the breast cancer cell line MDA-MB-231 stably overexpressing tGLI1. Ketoconazole activity against normal brain and breast microenvironmental cells. Note: *, *p* < 0.05; **, *p* < 0.01; ***, *p* < 0.001; ****, *p* < 0.0001; two-way (**b**,**c**) or one-way (**d**) ANOVA with post hoc Bonferroni’s or Dunnett’s multiple comparison test, respectively. The uncropped blots are shown in page 1 of Appendix A.

**Figure 2 cancers-14-04256-f002:**
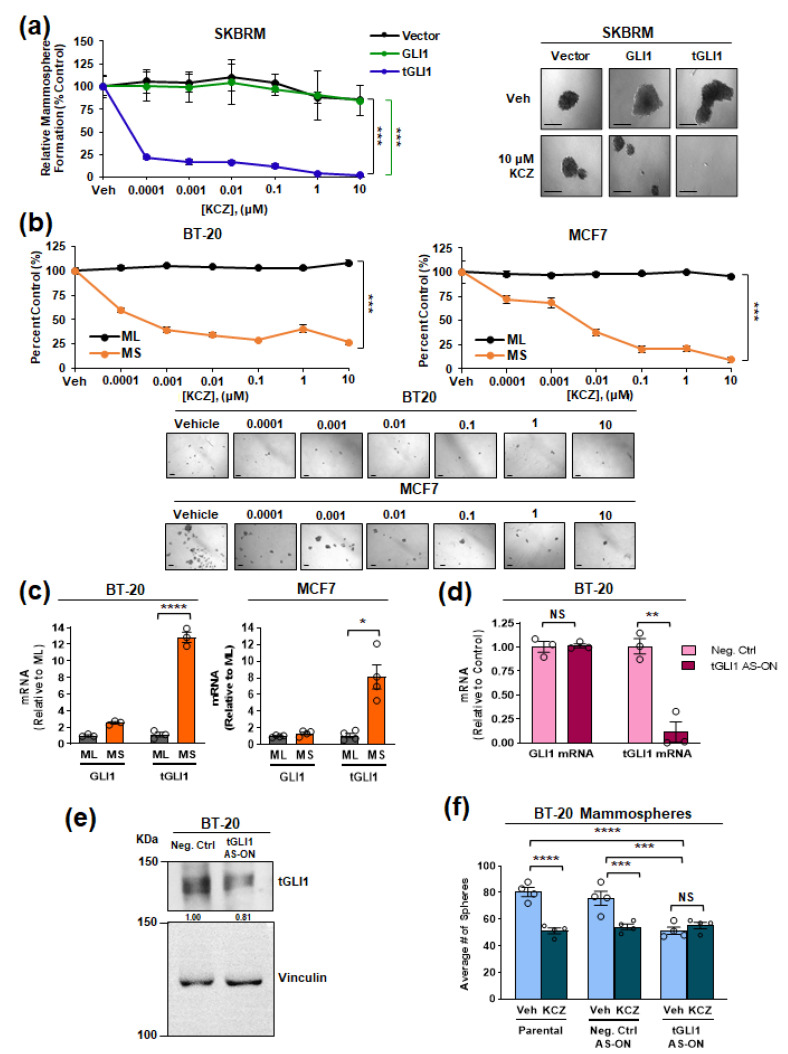
The tGLI1 expression is required for the KCZ-induced suppression of breast CSCs. (**a**) Mammosphere formation of SKBRM stable cell lines after 7 days of treatment (left). Representative images of mammospheres at day 7 (right). Scale bar represents 200 μm. (**b**) KCZ activity against parental breast CSCs expressing endogenous tGLI1 and the total cell population. Parental BT-20 (left) and MCF7 (right) were grown concurrently as mammospheres (MS, orange) or monolayers (ML, black) and treated with KCZ (top). ML viability was determined by CellTiter-Blue^®^ Viability assay while the number of spheres was used to assess the CSC subpopulation. Representative images of BT-20 or MCF-7 mammospheres after KCZ treatment (bottom). Scale bar represents 100 µm. (**c**) GLI1 and tGLI1 expression in BT-20 (left) and MCF7 (right) MS relative to the ML. (**d**) GLI1 and tGLI1 mRNA expression in BT-20 cells transfected with a tGLI1 antisense oligonucleotide (AS-ON) as indicated by qPCR. (**e**) The tGLI1 protein expression in BT-20 cells transfected with a tGLI1 ASON, as indicated by Western blotting. (**f**) The tGLI1-knockdown attenuates the KCZ-mediated inhibition of BT-20 mammosphere formation. Note: *, *p* < 0.05; **, *p* < 0.01; ***, *p* < 0.001; ****, *p* < 0.0001; two-way ANOVAs with post hoc Bonferroni’s multiple comparison test (**a**,**b**,**f**) and two-tailed Student’s *t*-test (**c**,**d**) were used to calculate *p*-values. The uncropped blots are shown in page 1 of Appendix A.

**Figure 3 cancers-14-04256-f003:**
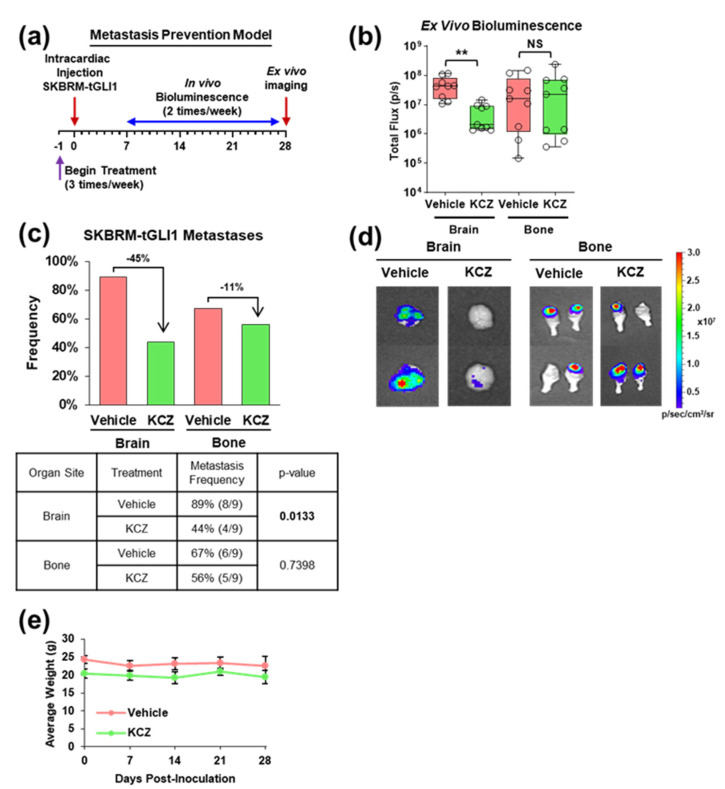
KCZ reduces the ability of the tGLI1-positive circulating breast cancer cells to undergo colonization and form brain metastases in vivo. (**a**) Schema for the intracardiac brain metastasis prevention model. Isogenic luciferase-expressing SKBRM-tGLI1 cells were injected into the left ventricle of female nude mice and the tumor growth was assessed biweekly via bioluminescent imaging. The mice received a single intraperitoneal treatment of 100 μL PEG-300 or 50 mg/kg KCZ 24 h prior to inoculation and continued to receive treatment 3 times per week (*N* = 9 per group). (**b**) Ex vivo brain and bone bioluminescence. (**c**) Incidence rates of brain and bone metastases in each treatment group at study endpoint. (**d**) Representative bioluminescent images of ex vivo brain and bone metastases. (**e**) Average weight of mice treated with vehicle or KCZ. Note: **, *p* < 0.01; two-tailed Student’s *t*-test (**b**) and an exact binomial test (**c**) were used to calculate *p*-values.

**Figure 4 cancers-14-04256-f004:**
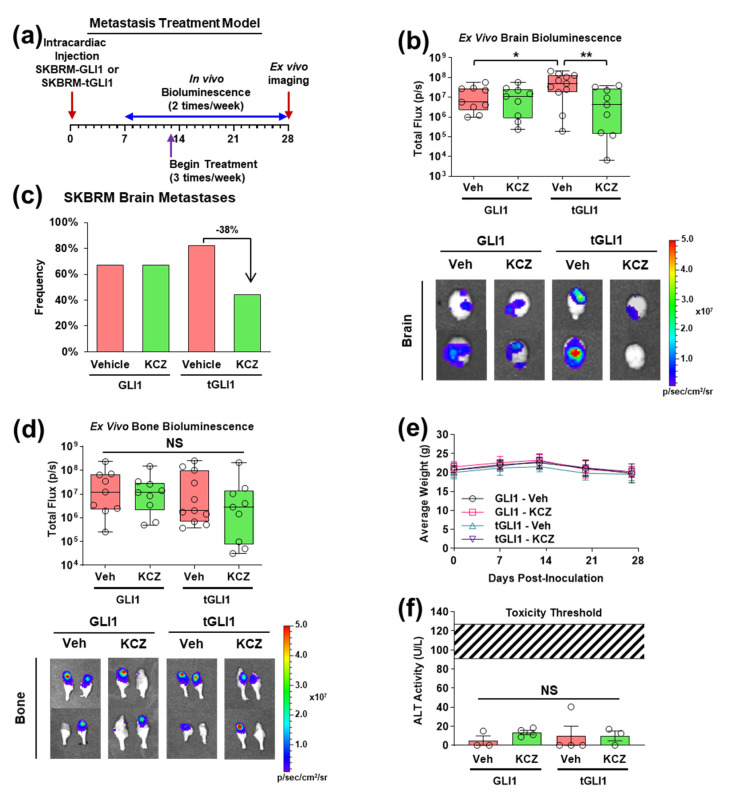
KCZ selectively reduces the progression of tGLI1-positive breast cancer brain metastases in vivo. (**a**) Schema for intracardiac brain metastasis treatment model. Isogenic luciferase-expressing GLI1 or tGLI1 SKBRM cells were injected into the left ventricle of female nude mice and tumor growth was assessed biweekly via bioluminescent imaging. Mice received 3 treatments per week of 100 μL PEG-300 or 50 mg/kg KCZ beginning 13 days post-inoculation (*N* = 9–10 per group). (**b**) Ex vivo brain bioluminescence at study endpoint (top). Representative ex vivo brain bioluminescence images (bottom). (**c**) Brain metastasis incidence at study endpoint. (**d**) Ex vivo bone bioluminescence at study endpoint (top). Representative ex vivo bone bioluminescence images (bottom). (**e**) Average weight of mice in each group. (**f**) Serum alanine transaminase (ALT) activity. Striped region represents range of ALT activity in athymic mice following thioacetamide-induced acute liver injury. Note: *, *p* < 0.05; **, *p* < 0.01; two-way ANOVA with post hoc Bonferroni’s multiple comparison test (**b**,**d**,**f**) was used to calculate *p*-values.

**Figure 5 cancers-14-04256-f005:**
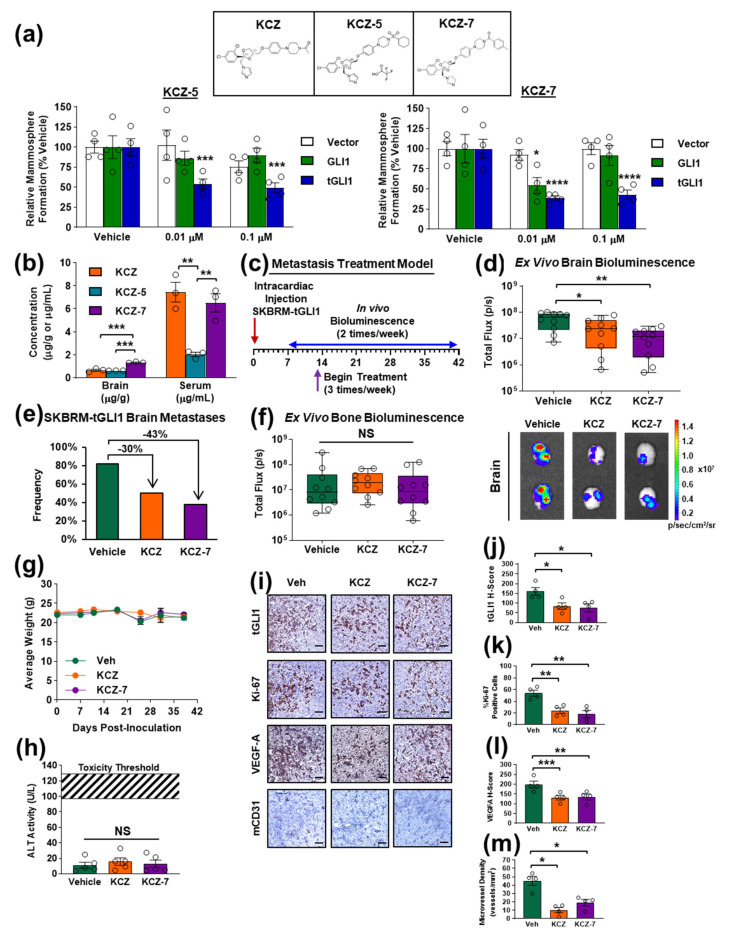
The novel KCZ derivative KCZ-7 retains its tGLI1 selectivity and in vivo efficacy while readily permeating the BBB. (**a**) The structures of KCZ, KCZ-5, and KCZ-7 (top). The effect of treatment with KCZ-5 (left) or KCZ-7 (right) on mammosphere formation of SKBRM cell lines. Asterisks denote comparison to intra-cell line vehicle treatment. (**b**) Concentrations of KCZ, KCZ-5, and KCZ-7 measured in whole mouse brain homogenate or mouse serum. Mice received a single intraperitoneal treatment of 50 mg/kg KCZ, KCZ-5, or KCZ-7. Matched whole blood and brain samples were collected 20 min after treatment and analyzed using mass spectrometry (*N* = 3). (**c**) Schema for intracardiac brain metastasis treatment model. (**d**) Ex vivo brain bioluminescence at study endpoint (top) (*N* = 10 per group). Representative ex vivo brain bioluminescent images (bottom). (**e**) Brain metastasis incidence rates at study endpoint. (**f**) Ex vivo bone bioluminescence at study endpoint (top) (*N* = 10 per group). (**g**) Average weight of mice in each group. (**h**) Serum alanine transaminase (ALT) activity. (**i**) Representative tGLI1, Ki-67, VEGF-A, and mCD31 expression levels as assessed by IHC in mice bearing SKBRM-tGLI1 brain metastases treated with vehicle, 50 mg/kg KCZ, or 50 mg/kg KCZ-7. (**j**–**m**) IHC quantification of tGLI1 expression (**j**), proliferative index (**k**), VEGF-A expression (**l**), and microvessel density (**m**) levels of SKBRM-tGLI1 brain metastases (*N* = 4 per group). Scale bar represents 100 μm. NS, not significant; *, *p* < 0.05; **, *p* < 0.01; ***, *p* < 0.001; ****, *p* < 0.0001; two-way ANOVAs with post hoc Dunnett’s multiple comparison test (**a**) or one-way ANOVA with post hoc Tukey’s multiple comparison test (**b**,**d**,**f**,**h**,**i**–**m**) were used to calculate *p*-values.

**Figure 6 cancers-14-04256-f006:**
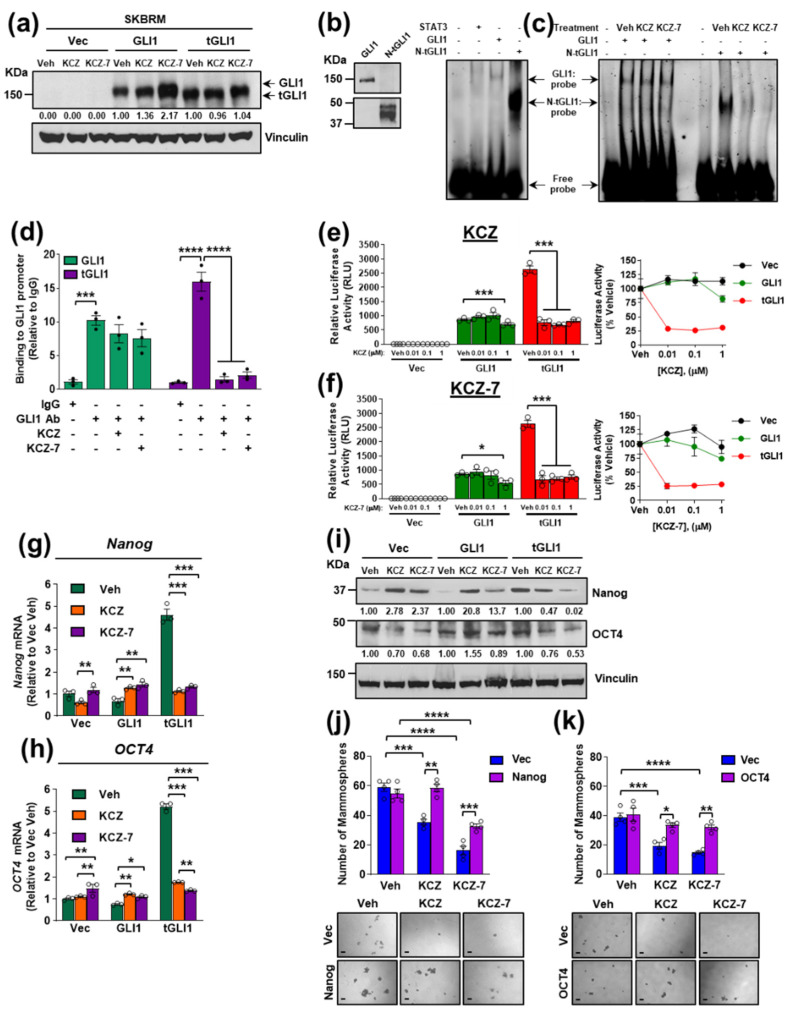
KCZ and the novel derivative KCZ-7 inhibit tGLI1 transcriptional activity leading to downregulation of validated tGLI1-mediated stemness genes *Nanog* and *OCT4*. (**a**) Representative Western blots of GLI1 and tGLI1 expression in isogenic SKBRM cell lines following 24 h treatment with vehicle, 1 μM KCZ, or 1 μM KCZ-7. The same membrane was probed to assess the loading control. (**b**) Western blots of recombinant GLI1 and N-tGLI1 (left). A tGLI1-selective Ab was used to detect tGLI1. Binding of recombinant GLI1 and N-tGLI1 to a dsDNA oligonucleotide containing the consensus GLI1/tGLI1-binding site (right). STAT3 was used as a negative control. (**c**) The DNA-binding ability of recombinant N-tGLI1, but not GLI1, is disrupted by KCZ or KCZ-7 treatment. (**d**) Relative binding of GLI1 or tGLI1 to the GLI1-binding sites in SKBRM cells, as determined by chromatin immunoprecipitation; qPCR was performed using primers spanning the GLI1 binding site. (**e**,**f**) Inhibition of GLI1- and tGLI1-mediated promoter transactivation by KCZ (**e**) and KCZ-7 (**f**). SKBR3 cells were transiently transfected with 8 × 3′GLI1 luciferase reporter and vector, GLI1, or tGLI1 plasmids, then treated with increasing doses of KCZ (**e**) or KCZ-7 (**f**) for 48 h and stimulated with SHH ligand (100 ng/mL) for 4 h. Right: Relative luciferase activity normalized to vehicle treatment. (**g**,**h**) Selective reduction of tGLI1-mediated stemness genes Nanog (**g**) and OCT4 (**h**) mRNA as assessed by RT-qPCR in isogenic SKBRM cell lines treated with vehicle, 1 μM KCZ, or 1 μM KCZ-7 for 24 h. (**i**) Nanog and OCT4 protein expression following treatment with vehicle, 1 μM KCZ, or 1 μM KCZ-7 in isogenic SKBRM cell lines. The same membrane was probed to assess the loading control. (**j**,**k**) Overexpression of Nanog (**j**) or OCT4 (**k**) rescues SKBRM-tGLI1 mammospheres from KCZ and KCZ-7 treatment. Scale bars represent 200 μm. N-tGLI1, N-terminal tGLI1; *, *p* < 0.05; **, *p* < 0.01; ***, *p* < 0.001; ****, *p* < 0.0001; two-way ANOVA with post hoc Dunnett’s (**d**–**f**) or Bonferroni’s (**g**,**h**,**j**,**k**) multiple comparison test was used to calculate *p*-values. The uncropped blots are shown in page 2 of Appendix A.

## Data Availability

The data generated by this study are available on request from the corresponding author.

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
