# Peer review of "An FDA-Approved Antifungal, Ketoconazole, and Its Novel Derivative Suppress tGLI1-Mediated Breast Cancer Brain Metastasis by Inhibiting the DNA-Binding Activity of Brain Metastasis-Promoting Transcription Factor tGLI1"

_cancers, 2022, doi:10.3390/cancers14174256_

Round 1

Reviewer 1 Report (Previous Reviewer 2)

Authors have completed all the concerns in the revised version which were  raised in previous version of the manuscript.  

I recommend this version of the manuscript for the publication, however, the Table S2 could have been represented in the excel file as the visibility for the data in this pdf file very poor. 

Reviewer 2 Report (Previous Reviewer 1)

The current version has addressed my concerns raised last time and is suitable for publication.

This manuscript is a resubmission of an earlier submission. The following is a list of the peer review reports and author responses from that submission.

Round 1

Reviewer 1 Report

Daniel Doheny et al applied the compounds library to identify the potential inhibitor of truncated glioma-associated oncogene homolog 1 (tGLI1), which was previously well-studied by the corresponding author’s lab. They finally identified that ketoconazole (KCZ), a FDA-approved antifungal drug, selectively targets tGLI1-positive breast cancer cells and breast cancer stem cells in vitro and in vivo. Overall, this is an interesting work and has a potential impact on clinical practice in the prevention and treatment of breast cancer brain metastatic. However, the current data presented is still premature.

1.      Western blot results of the exogenous overexpressed GLI1 and tGLI1 should be presented in Fig 1. Additionally, the expression of tGLI1 in human astrocytes, HBMEC, HMLE, and MCF10A should also be shown by WB assay whether it is positive or negative. The expression of tGLI1 in mRNA and protein level in MDA-MB-231 and MDA-MB-231BRM should be shown.

2.      In fig. 2D, Western blot results showing the reduced level of tGLI1 should be presented.

3.      In fig. 3C, the statistical analysis should be provided. 

4.      Why KCZ primarily inhibit the tumors colonized in brain not in lung and bones since in vitro results is clear that KCZ inhibited the growth and colony formation in tGLI1-positive cells. Did you compare the level of tGLI1 in brain tumor with that in lung and bone tumors?

5.      In fig. 5i, administration of both KCZ and KCZ-7 significantly reduced levels of tGLI1 in vivo. Why a similar effect can’t be found in in vitro assay as shown in fig. 6a?

Reviewer 2 Report

The manuscript submitted by Doheny et al, have screened a library of almost 1500 compounds and find ketoconazole (KCZ), and its novel derivative to target tGLI1 36 transcriptional activity, suppress cancer stem cells, and inhibit BCBM. This is an interesting topic to be explored at the current scenario and the manuscript is well written, and data presented nicely with proper control.  However, there are still some inaccuracies, and some experiments might be improved.

Line 280 Authors should present the screening results in supplementary files

Figure1a 1b, 1c why treated cells in vector are showing the higher cell viability as compared to vehicle treatment

Figure 2b, the respective images of ML and MS treated vs non treated should be presented either in main figure or supplementary

Figure 2c, Authors have reported that KCZ significantly reduced mammosphere formation of HER2-enriched tGLI1-expressing brain metastatic SKBR3 (SKBRM) cells, therefore, it would be advisable to check the effects of KCZ in tGLI1 expression in SKBR3 (SKBRM) cells.

Line 445, Authors should present the data showing the purity of the derivatives (NMR, Mass spectra etc.)

Line 448, Authors should also perform the cell viability of these derivatives in SKBR3

Line 509-Line 511, The statement confuses me, as if the compounds does not change the tGLI1 protein expression, how it is hypothesized that it interacts directly, please explain

Figure 6b, left panel, Authors should present the full images of protein purification in the supplementary
